# Comorbidities and Outcome of Alcoholic and Non-Alcoholic Liver Cirrhosis in Taiwan: A Population-Based Study

**DOI:** 10.3390/ijerph17082825

**Published:** 2020-04-20

**Authors:** Tzu-Wei Yang, Chi-Chih Wang, Ming-Chang Tsai, Yao-Tung Wang, Ming-Hseng Tseng, Chun-Che Lin

**Affiliations:** 1School of Medicine, Chung Shan Medical University, Taichung 402, Taiwan; joviyoung@gmail.com (T.-W.Y.); bananaudwang@gmail.com (C.-C.W.); tmc1110@yahoo.com.tw (M.-C.T.); wangyt@ms9.hinet.net (Y.-T.W.); 2Institute of Medicine, Chung Shan Medical University, Taichung 402, Taiwan; 3Division of Gastroenterology and Hepatology, Department of Internal Medicine, Chung Shan Medical University Hospital, Taichung 402, Taiwan; 4Division of Pulmonary Medicine, Department of Internal Medicine, Chung Shan Medical University Hospital, Taichung 402, Taiwan; 5Department of Medical Informatics, Chung Shan Medical University, Taichung 402, Taiwan; 6Information Technology Office, Chung Shan Medical University Hospital, Taichung 402, Taiwan; 7Department of Internal Medicine, China Medical University Hospital, Taichung 404, Taiwan; 8School of Medicine, China Medical University, Taichung 404, Taiwan

**Keywords:** alcoholic liver cirrhosis, non-alcoholic liver cirrhosis, survival, comorbidity

## Abstract

The prognosis of different etiologies of liver cirrhosis (LC) is not well understood. Previous studies performed on alcoholic LC-dominated cohorts have demonstrated a few conflicting results. We aimed to compare the outcome and the effect of comorbidities on survival between alcoholic and non-alcoholic LC in a viral hepatitis-dominated LC cohort. We identified newly diagnosed alcoholic and non-alcoholic LC patients, aged ≥40 years old, between 2006 and 2011, by using the Longitudinal Health Insurance Database. The hazard ratios (HRs) were calculated using the Cox proportional hazards model and the Kaplan–Meier method. A total of 472 alcoholic LC and 4313 non-alcoholic LC patients were identified in our study cohort. We found that alcoholic LC patients were predominantly male (94.7% of alcoholic LC and 62.6% of non-alcoholic LC patients were male) and younger (78.8% of alcoholic LC and 37.4% of non-alcoholic LC patients were less than 60 years old) compared with non-alcoholic LC patients. Non-alcoholic LC patients had a higher rate of concomitant comorbidities than alcoholic LC patients (79.6% vs. 68.6%, *p* < 0.001). LC patients with chronic kidney disease demonstrated the highest adjusted HRs of 2.762 in alcoholic LC and 1.751 in non-alcoholic LC (all *p* < 0.001). In contrast, LC patients with hypertension and hyperlipidemia had a decreased risk of mortality. The six-year survival rates showed no difference between both study groups (*p* = 0.312). In conclusion, alcoholic LC patients were younger and had lower rates of concomitant comorbidities compared with non-alcoholic LC patients. However, all-cause mortality was not different between alcoholic and non-alcoholic LC patients.

## 1. Introduction

Liver cirrhosis (LC) resulting from different etiologies is a leading cause of death, which accounts for 3.5% of all death worldwide [1,2]. The leading cause of LC is alcohol-related liver disease in developed countries, which is responsible for 0.9% of all global deaths in 2010 [1,3]. Central Asia had the highest number of alcoholic LC-related deaths with 17.5 deaths per 100,000 people followed by Central Latin America in 2010 [4]. On the other hand, infection with viral hepatitis is the most common cause of LC in endemic areas such as Asia and Saharan Africa [3]. Two population-based cohort studies in England reported a higher all-cause mortality rate in alcoholic LC than in non-alcoholic LC [5,6]. However, the majority of the study subjects had alcohol-related liver disease. The comparison of clinical outcome between alcoholic LC and non-alcoholic LC patients is not well understood.

Given that the survival of patients with LC is determined by liver function, worsening of liver function is largely attributed to concomitant comorbidities and complications of LC [7,8,9]. Several prediction models for mortality have been developed for end-stage liver disease (e.g., Charlson comorbidity index (CCI), model for end-stage liver disease (MELD), and five-variable MELD (5vMELD)) [10,11,12]. However, most of these survival prediction models were developed using alcoholic liver disease-dominated cohorts. To date, the comparison of clinical outcome between alcoholic and non-alcoholic liver disease in viral hepatitis-dominated cohort is not well understood. Taiwan is one of the endemic areas for viral hepatitis. The prevalence rates of hepatitis B and hepatitis C were 13.7% and 4.4% in the general population, respectively [13,14]. The majority of the cause of liver disease and liver disease-related mortality in patients was viral hepatitis [15]. The non-alcoholic LC-dominated cohort in Taiwan can provide a better understanding of the outcome.

Herein, we conducted a population-based cohort study to compare the survival between alcoholic and non-alcoholic LC. We identified newly diagnosed LC patients by using the nationwide registration database, followed the survival, and analyzed the risk of concomitant comorbidities and complications.

## 2. Methods

### 2.1. Study Population and Design

We conducted a nationwide retrospective longitudinal population-based cohort study by using the Taiwanese National Health Insurance Research Database (NHIRD). The NHIRD provides coverage for 99% of the Taiwan population since 1 March 1995. To select the study subjects in the cohort, we identified all LC patients in the admission and outpatient files. The first-time diagnosis of LC was used as the index date. To identify patients with alcoholic and non-alcoholic LC from 2004 to 2011, we used the International Classification of Disease, 9th Revision, Clinical Modification (ICD-9-CM) codes 571.2 and 571.5, respectively. To exclude overlapping patients, subjects with concomitant ICD-9-CM codes 571.5 and 571.6 (biliary LC) were excluded in alcoholic LC groups and subjects with concomitant ICD-9-CM codes 571.2 and 571.6 were excluded in non-alcoholic LC groups. Other exclusion criteria were patients who had LC before the end of 2005, patients aged <40 years on the index date, and outpatient follow-ups with the same diagnosis less than two times after the index date (Figure 1). Urbanization was categorized as metropolis, general area, and remote area according to residential areas. The socioeconomic status was divided into three levels according to monthly income as low income (0–14,009 New Taiwan Dollar (NTD)), mid income (14,010–42,030 NTD), and high income (>42,939 NTD).

### 2.2. Comorbidities and Complications

The International Classification of Disease, Ninth Revision, Clinical Modification (ICD-9-CM) diagnosis codes used to identify comorbidities and complications of LC data are summarized in Appendix A. The diagnosis of comorbidities was defined as having three outpatient visits or one admission. The diagnosis of complications was identified in the admission file. Esophageal varices (EV) with bleeding was confirmed by the ICD-9 procedure code of ligation of esophageal varices or sclerotherapy.

### 2.3. Study Endpoint Measurement

We observed the study subjects with a follow-up duration of six years. The study endpoint was to compare the all-cause mortality between alcoholic and non-alcoholic LC groups. We also used the Cox proportional hazards model to analysis the concomitant comorbidities and complications of LC.

### 2.4. Statistical and Data Analysis

To process the data from the NHIRD database, we used Microsoft SQL Server 2008 R2 (Microsoft Corporation, Redmond, WA, USA). All data were managed with the SPSS software 19.0 (SPSS, Inc., Chicago, IL, USA). We used the Chi-square test and analysis of variance (ANOVA) to analyze the demographic data, concomitant comorbidities, and complications of cirrhosis. We used the Kaplan–Meier plot and Cox proportional hazards model to calculate the all-cause mortality hazard ratios (HRs) of comorbidities and complications of LC in all study subjects. The results are expressed in unadjusted and adjusted hazard ratios with 95% confidence intervals. The six-year cumulative survival plot was calculated using the Cox regression model. A two-tailed *p*-value less than or equal to 0.05 was considered to be statistically significant.

### 2.5. Ethics

This study was approved by the Institutional Review Board (IRB) of Chung Shan Medical University Hospital, Taiwan (CSMUH No: CS13007) on January 10, 2013. The IRB waived the need for informed consent for this retrospective study based on an encrypted National Health Insurance Research Database (NHIRD).

## 3. Results

A total of 4785 liver cirrhosis patients, including 472 alcoholic LC and 4313 non-alcoholic LC, were identified. Table 1 lists the detailed demographic characteristics and comorbidities of the two cohorts. Male gender accounted for higher proportions in both groups, especially in the alcoholic LC cohort (94.7% of patients were male). In contrast to non-alcoholic LC patients, alcoholic LC patients were younger (*p* < 0.001). Nearly 80% of alcoholic LC patients were diagnosed at the age of 40–59 years. The residential urbanization level was different among both groups, with a smaller number of patients from remote area compared to metropolis and general area in both the groups. Compared with non-alcoholic LC patients, more than half of the alcoholic patients had mid income (Table 1).

### 3.1. Concomitant Comorbidities and Complications of Liver Cirrhosis

The prevalence of concomitant comorbidity was lower in alcoholic LC than in non-alcoholic LC (68.6% vs. 79.6%, *p* < 0.001). Coronary artery disease (CAD), hemorrhagic and ischemic stroke, hypertension, heart failure (HF), diabetes mellitus (DM), chronic kidney disease (CKD), and chronic obstructive pulmonary disease (COPD) were more prevalent in non-alcoholic patients (all *p* < 0.001) (Table 1). The comparison of complications of LC between the two groups showed that the non-alcoholic LC group was associated with a higher rate of any complication of LC (*p* = 0.007), especially ascites or peritonitis (23.5% vs. 16.7%, *p* < 0.001) and esophageal varices without bleeding (9.2% vs. 5.3%, *p* = 0.005) (Table 1).

### 3.2. Hazard Ratios for Comorbidities, Complications of Liver Cirrhosis, and Survival

We performed a Cox proportional hazards regression analysis of comorbidities and complications of LC by using all-cause mortality as the endpoint. Male gender was associated with poor outcome in the non-alcoholic LC group and the adjusted HRs (aHRs) were 1.241 (95% CI = 1.119–1.376) and 1.363 (95% CI = 1.231–1.510) when analyzing comorbidities and complications, respectively (Table 2, Appendix A). A higher risk of mortality was also measured in older patients (>69 years old). Among the comorbidities, chronic kidney disease (CKD) had the highest aHRs at 2.276 (95% CI = 1.520–3.409) in the alcoholic LC group and 1.761 (95% CI = 1.571–1.951) in the non-alcoholic LC group. Non-alcoholic LC patients with hemorrhagic stroke were also associated with a higher aHR of mortality (1.377, 95% CI = 1.090–1.738). Of note, LC patients with hypertension were associated with lower risks of death in the alcoholic group (aHR = 0.684, 95% CI = 0.471–0.994) and non-alcoholic group (aHR = 0.783, 95% CI = 0.698–0.878), as well as hyperlipidemia in the alcoholic group (aHR = 0.447, 95% CI = 0.282–0.708) and non-alcoholic group (aHR = 0.571, 95% CI = 0.499–0.652). 

When the complications of LC were compared, alcoholic LC patients with hepatic encephalopathy (aHR = 2.231, 95% CI = 1.449–3.436) and esophageal varices (aHR = 1.853, 95% CI = 1.231–2.788) exhibited a higher risk of mortality. In the non-alcoholic LC group, ascites or peritonitis exhibited the highest aHR at 2.052 (95% CI = 1.841–2.288), followed by hepatic encephalopathy (aHR = 1.971, 95% CI = 1.745–2.225) (Appendix A).

### 3.3. Survival Analysis Between the Two Study Groups

The cumulative survival during the follow-up period is plotted in Figure 2 by using the Cox proportional hazards model for alcoholic and non-alcoholic LC patients.

The HR of alcoholic LC was 0.919 (95% CI = 0.781–1.082) compared to the non-alcoholic LC group and was not statistically significant.

## 4. Discussion

### 4.1. Main Findings

In the study, we compared alcoholic LC and non-alcoholic LC patients. We observed that individuals with alcoholic LC were predominantly male and younger, had low or mid income, and had a lower prevalence of comorbidities compared to non-alcoholic LC patients. However, the six-year survival rates after diagnosis were not different between alcoholic and non-alcoholic LC patients. Alcoholic LC patients with CKD, hepatic encephalopathy, or esophageal varices (EV) had a higher risk of mortality. Non-alcoholic LC patients with hemorrhagic stroke, CKD, ascites or peritonitis, or EV had a higher risk of mortality. We also found that patients with concomitant hypertension and hyperlipidemia were associated with a lower risk of mortality in both alcoholic LC and non-alcoholic LC groups.

### 4.2. Comparison with Other Studies

Previous studies on alcoholic liver disease (ALD) reported that ALD-related mortality was more prevalent in the male gender and mostly occurred in the age of 35–64 years [4,16,17,18]. In our alcoholic LC group, male gender was significantly more prevalent than female (male:female = 94.7%:5.2%). Interestingly, the aHRs for male gender after adjustment for comorbidities and complications were 0.777 (95% CI = 0.399–1.512) and 0.855 (95% CI = 0.444–1.647), respectively, which showed no statistical difference in alcoholic patients (Table 2, Appendix A). The aHR of age was significantly higher in alcoholic patients who were more than 69 years old and non-alcoholic LC patients who were more than 59 years old.

A population-based study in China demonstrated a younger onset of ALD than in western countries [17,18]. Most patients diagnosed with ALD were 40–49 years old [19]. In the present study, we found that most patients with a diagnosis of alcoholic LC were 40–49 years old, and four-fifths of the patients were diagnosed before they were 60 years old. This finding is probably due to the lower aldehyde dehydrogenase 2 activity, which is more susceptible to alcohol-induced liver injury in Asian and Chinese populations [20,21,22].

Compared with the non-alcoholic group, more male patients, living in metropolis and from a low-income family, were found in the alcoholic LC group. This result is compatible with previous reports [19,23].

A previous population-based cohort study reported that alcoholic LC had a higher risk of all-cause mortality compared with viral LC, especially liver- and circulatory-related death [5]. However, circulatory disease-related comorbidities, such as hypertension and DM, were less prevalent in our alcoholic LC cohort. The lower prevalence of these aging-related comorbidities could be attributed to the young age of patients of this study cohort. In addition, all-cause mortality showed no difference between alcohol and non-alcoholic LC in our study.

Of note, hypertension carried a lower risk for mortality in both cohorts, which may be due to the improvement of hemodynamic defects (i.e., low systemic vascular resistance and low renal blood flow) in cirrhotic patients [24].

Among the comorbidities of Danish cirrhosis patients, metastatic cancer (HR = 1.72, 95% CI = 1.53–1.94) exhibited the highest HR, followed by CKD (HR = 1.59, 95% CI = 1.37–1.83) and acute myocardial infarction (HR = 1.59, 95% CI = 1.40–1.79) [9]. In the present study, we found that CKD exhibited the highest aHR in both the alcoholic LC (HR = 2.276, 95% CI = 1.520–3.409) and non-alcoholic LC cohorts (HR = 1.751, 95% CI = 1.571–1.951).

### 4.3. Clinical Implications

In alcoholic LC patients, hepatic encephalopathy (HE) is the most common admission diagnosis [16]. We found that HE had the highest aHR among the complications of alcoholic LC. Therefore, we should emphasize the prevention and management of HE in alcoholic LC patients. On the other hand, the highest aHR in non-alcoholic LC patients was observed for ascites and peritonitis. The aggressive management of ascites by fluid or dietary sodium restriction and antibiotic prophylaxis for non-alcoholic LC patients with risk factors for spontaneous bacterial peritonitis may have some favorable impact on cirrhosis mortality.

### 4.4. Strengths and Limitations

The strength of the present study is the population-based database cohort. In the two study groups, we excluded overlapping subjects before analysis. The first important limitation of the study was liver function. The median survival time of compensated LC patients has been reported to be six times longer than that in decompensated LC patients [7]. We could not categorize the patients by severity because of the limitation of the NHIRD (i.e., the Child–Turcotte–Pugh score and MELD score are not available in the NHIRD). Second, we did not classify the different etiologies in the non-alcoholic LC group, including viral hepatitis, non-alcoholic fatty liver disease, and autoimmune hepatitis. The different etiologies of liver disease are associated with the different development and progression of liver fibrosis and LC. Third, given that excessive alcohol consumption is associated with hyperlipidemia [25,26], the amount, timing, and type of alcohol consumption play important roles in the development of hyperlipidemia and metabolic syndrome [27]. However, the amount of alcohol consumption could not be evaluated in our study cohort due to dataset limitation. Finally, we did not analyze the cause of death due to the limitations of the dataset.

## 5. Conclusions

In summary, alcoholic LC patients were younger and had lower rates of concomitant comorbidities. The six-year survival rates after diagnosis were not different between alcoholic and non-alcoholic LC. Both alcoholic and non-alcoholic LC patients with concomitant hypertension and hyperlipidemia were associated with a lower risk of mortality.

## Figures and Tables

**Figure 1 ijerph-17-02825-f001:**
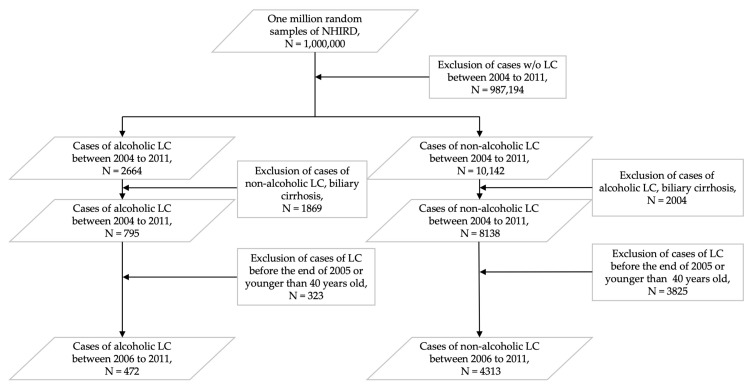
Flowchart of enrollment of alcoholic liver cirrhosis (LC) and non-alcoholic LC in the study cohort. Abbreviations: LC: liver cirrhosis; NHIRD: National Health Insurance Research Database; w/o: without. A total of 323 patients, including patients with LC diagnosed before 2005 (N = 270) and aged <40 years (N = 53), were excluded from the alcoholic LC cohort between 2004 to 2011. In the non-alcoholic LC cohort between 2004 to 2011, 3825 patients, including patients with LC diagnosed before 2005 (N = 3608) and aged <40 years (N = 217), were excluded.

**Figure 2 ijerph-17-02825-f002:**
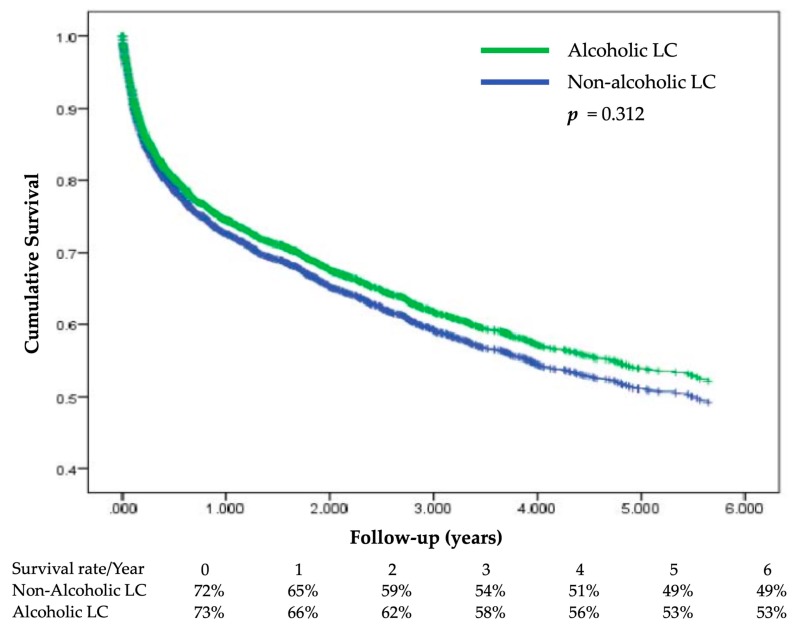
Cumulative survival plot using the Cox model for alcoholic and non-alcoholic LC patients during the follow-up period. After Cox regression, the hazard ratio of alcoholic LC was 0.919 (95% CI = 0.781–1.082, *p* = 0.312) compared to the non-alcoholic LC group.

**Table 1 ijerph-17-02825-t001:** Baseline characteristics of study subjects.

Variable	Alcoholic Cirrhosis		Non-Alcoholic Cirrhosis		*p*-Value
	N = 472		N = 4313		
	*N*	%	*N*	%	
Sex					<0.001
Male	447	94.7	2699	62.6	
Female	25	5.3	1614	37.4	
Age (years)					<0.001
40–49	220	46.6	601	13.9	
50–59	152	32.2	1015	23.5	
60–69	68	14.4	974	22.6	
>69	32	6.8	1723	39.9	
Residence					<0.001
Metropolis	246	52.1	2403	55.7	
General area	205	43.4	1837	42.6	
Remote area	20	4.2	68	1.6	
Income (New Taiwan Dollar (NTD) per month)					<0.001
0–14,009	201	42.6	2175	50.4	
14,010-42,030	248	52.5	1838	42.6	
>42,939	22	4.7	295	6.8	
Etiologies of non-alcoholic cirrhosis					
Hepatitis B			1417	32.9	
Hepatitis C			1377	31.9	
Non-alcoholic fatty liver disease			428	9.9	
Others (biliary cirrhosis excluded)			1091	25.3	
Comorbidity					<0.001
Yes	324	68.6	3431	79.6	
No	148	31.4	882	20.4	
Coronary heart disease (CAD)	63	13.3	1091	25.3	<0.001
Cerebrovascular disease					
Hemorrhage	15	3.2	578	13.4	<0.001
Ischemia	24	5.1	684	15.9	<0.001
Hypertension	197	41.7	2526	58.6	<0.001
Heart failure (HF)	25	5.3	551	12.8	<0.001
Diabetes mellitus (DM)	137	29.0	1584	36.7	<0.001
Chronic kidney disease (CKD)	72	15.3	970	22.5	<0.001
Hyperlipidemia	118	25.0	1035	24.0	0.629
Chronic obstructive pulmonary disease (COPD)	59	12.5	903	20.9	<0.001
Complication of cirrhosis					0.007
Yes	152	32.2	1663	38.6	
No	320	67.8	2650	61.4	
Ascites or peritonitis	79	16.7	1012	23.5	<0.001
Hepatic encephalopathy	54	11.4	553	12.8	0.392
Esophageal varices	57	12.1	656	15.2	0.069
Esophageal varices without bleeding	25	5.3	396	9.2	0.005
Esophageal varices with bleeding	45	9.5	456	10.6	0.484

**Table 2 ijerph-17-02825-t002:** Multivariable Cox proportional hazards model for comorbidities and survival analysis.

Variable	Alcoholic Cirrhosis						Non-Alcoholic Cirrhosis					
	Unadjusted Hazard Ratio			^a^ Adjusted Hazard Ratio			Unadjusted Hazard Ratio			^a^ Adjusted Hazard Ratio		
	Risk Ratio	95% CI	*p*-Value	Risk Ratio	95% CI	*p*-Value	Risk Ratio	95% CI	*p*-Value	Risk Ratio	95% CI	*p*-Value
Sex												
Male	0.703	0.371–1.335	0.282	0.777	0.399–1.512	0.458	1.114	1.008–1.232	0.034	1.241	1.119-1.376	<0.001
Female (reference)												
Age (years)												
40–49 (reference)												
50–59	1.255	0.877–1.795	0.214	1.335	0.923–1.930	0.124	1.070	0.883–1.297	0.489	1.114	0.918–1.351	0.275
60–69	0.793	0.465–1.353	0.395	0.848	0.489–1.472	0.559	1.258	1.041–1.521	0.017	1.344	1.105–1.635	0.003
>69	2.541	1.540–4.193	<0.001	2.004	1.115–3.603	0.020	2.487	2.104–2.939	<0.001	2.675	2.229–3.212	<0.001
Comorbidity												
Coronary heart disease	1.098	0.700–1.723	0.685	1.170	0.703-1.948	0.546	1.129	1.014-1.257	0.027	0.886	0.785-1.000	0.050
Cerebrovascular disease												
Hemorrhage	1.799	0.883–3.666	0.106	1.578	0.732–3.400	0.244	1.356	1.077–1.706	0.01	1.377	1.090–1.738	0.007
Ischemia	1.279	0.692–2.361	0.432	0.879	0.439–1.758	0.715	1.221	1.071–1.393	0.003	0.950	0.827–1.092	0.470
Hypertension	0.845	0.614–1.162	0.300	0.684	0.471–0.994	0.046	1.059	0.960–1.167	0.253	0.783	0.698–0.878	<0.001
Heart failure	1.393	0.754–2.571	0.290	0.981	0.477–2.016	0.958	1.510	1.328–1.715	<0.001	1.136	0.987–1.308	0.076
Diabetes mellitus	0.958	0.683–1.343	0.802	1.056	0.732–1.524	0.771	1.047	0.949–1.156	0.360	1.090	0.981–1.212	0.110
Chronic kidney disease	2.137	1.488–3.070	<0.001	2.276	1.520–3.409	<0.001	1.881	1.698–2.084	<0.001	1.751	1.571–1.951	<0.001
Hyperlipidemia	0.452	0.293–0.699	<0.001	0.447	0.282–0.708	<0.001	0.574	0.505–0.653	<0.001	0.571	0.499–0.652	<0.001
Chronic obstructive pulmonary disease	1.227	0.800-1.883	0.348	1.240	0.789-1.951	0.351	1.346	1.205-1.502	<0.001	0.994	0.883-1.119	0.920

^a^ Adjustments were made for sex, age, residence, income, coronary artery disease, cerebrovascular disease, hypertension, heart failure, diabetes mellitus, chronic kidney disease, hyperlipidemia, and chronic obstructive pulmonary disease.

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
