# Peer review of "Comorbidities and Outcome of Alcoholic and Non-Alcoholic Liver Cirrhosis in Taiwan: A Population-Based Study"

_ijerph, 2020, doi:10.3390/ijerph17082825_

Round 1
Reviewer 1 Report
Summary
The authors provide population-based data from Taiwan on patients with alcoholic versus non-alcoholic liver cirrhosis with detailed statistics on demographics, comorbidities, complications and all-cause mortality. The sample sizes are relevant for the evaluation and calculations, data are well presented and clinically relevant. Besides minor issues, I see only one major issue for improvement:
Given the data are available, it would be much more informative to perform a sex-specific analysis – some of the differences could results from the very different ratio of males to females in both groups. The female alcoholic LC group might be too small to draw conclusions, but it would be important to compare the male alcoholic LC to male non-alcoholic LC, and then it would be interesting to look at male versus female non-alcoholic LC, those groups are sufficiently large.
Details for minor issues:
Abstract
“We found that alcoholic LC patients were male predominant (94.7% male)” – it would be helpful to include the % of males in the non-alcoholic LC group for comparison.
Methods
The figure text is very small, please prepare a figure with larger text fields.
Please provide excluded cases separately for LC before 2005 and for younger than 40 years old.
What was the reason to first include patients with LC from 2004 to 2011 but then exclude those with LC before the end of 2005?
Results
“In contrast to non-alcoholic LC patients, alcoholic LC patients were at younger ages, especially in the age groups of 40-49 and 50-59 (P < .001).”
What is meant by “especially in the age groups …”? That these were the dominant ager groups? (It cannot be meant, that patients were younger especially in these age groups.) How was this p-value calculated, what does it mean? The same as in the table? Please rephrase to make it clear.
“Besides, more than half of the alcoholic patients had a mid-income (table 1).” Compared to non-alcoholic LC…? Otherwise the sentence is not very informative.
In “3.1. Concomitant Comorbidities and Complications of LC”, please spell out abbreviations CAD, COPD etc. at first use. Since they are very common, it might be helpful to include them (additionally in brackets) in table 1 for quicker reading of the information.
S2 supplementary table could be included in table 1 to have all information pooled in one table.
Table 2 – the lines with “No (reference)” for comorbidities should be removed, this could be stated in one sentence in the legend, if judged needed. With some reformatting, the entire table (incl. S3) could fit on one page. (Avoid line breaks for CI.) In all tables it would be helpful to mark significant differences (p<0.05) in bold.
“when analyzing comorbidities and complications respectively (table 2 and supplementary table S2)” – this should be supplementary table S3 not S2. Why not also include this information in table 2 to have one complete overview?
Discussion
“Interestingly, the aHR for comorbidities and complications showed no difference between both gender in alcoholic patients”
These data are not shown. The entire analysis should be performed separately for male and female patients in both groups showing all data for both sexes.
4.2. “Besides, we found male gender, living in metropolis, and low family income was associated with higher rate alcoholic LC …”
There are only % data in table 1 for living area and income but no statistics whether these factors have a significant effect (as they are not included in table 2). So it’s not clear.
4.3. “We found that HE had the highest AHR” should be aHR
4.3. “We found that HE had the highest AHR among the complications of LC.” But aHR for Ascites/Peritonitis in non-alcoholic LC is very high too, even higher than HE in this group. Please add and discuss as well.
Author Response
We thank the editor and reviewers for the opportunity to revise manuscript (ID: ijerph-759609) entitled: “Comorbidities and Outcome of Alcoholic and Non-alcoholic Liver Cirrhosis in Taiwan – A Population-based Study." The manuscript has been corrected in accordance with the reviewer's comments. All of the reviewers’ and editors’ comments were included and responded point-by-point. The below responses were colored, and the changes made in the text and the tables are highlighted in red.
Response to Reviewer #1:
Summary
The authors provide population-based data from Taiwan on patients with alcoholic versus non-alcoholic liver cirrhosis with detailed statistics on demographics, comorbidities, complications and all-cause mortality. The sample sizes are relevant for the evaluation and calculations, data are well presented and clinically relevant. Besides minor issues, I see only one major issue for improvement:
Given the data are available, it would be much more informative to perform a sex-specific analysis – some of the differences could results from the very different ratio of males to females in both groups. The female alcoholic LC group might be too small to draw conclusions, but it would be important to compare the male alcoholic LC to male non-alcoholic LC, and then it would be interesting to look at male versus female non-alcoholic LC, those groups are sufficiently large.
REPLY: We sincerely appreciate the positive feedback and suggestion. The study analyzing the non-alcoholic LC cohort had been published in our previous work (Tsai MC et al. World Journal of Gastroenterology 2018). We conducted further analysis to compare the comorbidities between male alcoholic LC and male non-alcoholic LC. The distribution of comorbidities is similar to that of the overall cohort (table attached to the last page). Please allow us to keep the full cohort for analysis.
Details for minor issues:
Abstract
“We found that alcoholic LC patients were male predominant (94.7% male)” – it would be helpful to include the % of males in the non-alcoholic LC group for comparison.
REPLY: We sincerely appreciate the positive feedback and suggestion. We rephrase the sentence as follows:
We found that alcoholic LC patients were male predominant (94.7% of alcoholic LC and 62.6% of non-alcoholic LC patients were male)
Methods
The figure text is very small, please prepare a figure with larger text fields.
REPLY: Thank you for the suggestion. We change the figure size with larger ones.
Please provide excluded cases separately for LC before 2005 and for younger than 40 years old.
REPLY: We add the data in the legend of table 1. Thank you.
A total of 323 patients, including LC diagnosed before 2005 (N=270) and age <40 years (N=53) were excluded from the alcoholic LC cohort between 2004 to 2011. In the non-alcoholic LC cohort between 2004 to 2011, 3,825 patients, including LC diagnosed before 2005 (N=3,608) and age <40 years (N=217) were excluded.
What was the reason to first include patients with LC from 2004 to 2011 but then exclude those with LC before the end of 2005?
REPLY: We aimed to analyze the newly diagnosed LC cases. Therefore, we used 2 years of washout period to make sure that patients did not receive a diagnosis of LC before the index date.
Results
“In contrast to non-alcoholic LC patients, alcoholic LC patients were at younger ages, especially in the age groups of 40-49 and 50-59 (P < .001).”
What is meant by “especially in the age groups …”? That these were the dominant ager groups? (It cannot be meant, that patients were younger especially in these age groups.) How was this p-value calculated, what does it mean? The same as in the table? Please rephrase to make it clear.
REPLY: We are sorry to have confused you. We rephrase the sentence as follows:
“In contrast to non-alcoholic LC patients, alcoholic LC patients were at younger ages (P < .001). Nearly 80% of alcoholic LC patients were diagnosed at the age of 40-59 years.”
“Besides, more than half of the alcoholic patients had a mid-income (table 1).” Compared to non-alcoholic LC…? Otherwise the sentence is not very informative.
REPLY: Thank you for the suggestion. We rephrase the sentence as follows:
“Besides, compared to non-alcoholic LC more than half of the alcoholic patients had a mid-income (table 1)”
In “3.1. Concomitant Comorbidities and Complications of LC”, please spell out abbreviations CAD, COPD etc. at first use. Since they are very common, it might be helpful to include them (additionally in brackets) in table 1 for quicker reading of the information.
REPLY: We sincerely appreciate the positive feedback and suggestion. We spell out the abbreviations in paragraph 3.1 and add the abbreviations in table 1.
S2 supplementary table could be included in table 1 to have all information pooled in one table.
REPLY: Thank you for the suggestion. We delete table S2 and include table S2 in table 1.
Table 2 – the lines with “No (reference)” for comorbidities should be removed, this could be stated in one sentence in the legend, if judged needed. With some reformatting, the entire table (incl. S3) could fit on one page. (Avoid line breaks for CI.) In all tables it would be helpful to mark significant differences (p<0.05) in bold.
REPLY: We sincerely appreciate the positive feedback and suggestion. We remove all the No (reference) in stable 2 and table S3. We also mark all the p<0.05 in bold in all tables.
“when analyzing comorbidities and complications respectively (table 2 and supplementary table S2)” – this should be supplementary table S3 not S2. Why not also include this information in table 2 to have one complete overview?
REPLY: We sincerely appreciate the positive feedback and suggestion. We correct the error. The factors for adjustments are different between table 2 and table S2. Please allow us to keep this information in separate tables.
Discussion
“Interestingly, the aHR for comorbidities and complications showed no difference between both gender in alcoholic patients”
These data are not shown. The entire analysis should be performed separately for male and female patients in both groups showing all data for both sexes.
REPLY: We sincerely appreciate the positive feedback and suggestion. We used female patients as reference to analyze the HR for male in table 2 and table S2. In stable 2, the aHR for mortality after adjusted by comorbidities was 0.777 (95 % CI = 0.399-1.512). In table S2, he aHR for mortality after adjusted by complications of LC was 0.855 (95 % CI = 0.444-1.647).
We rephrase the sentence as follows:
“Interestingly, the aHR for male gender after adjusted by comorbidities and complications were 0.777 (95 % CI = 0.399-1.512) and 0.855 (95 % CI = 0.444-1.647), respectively, which showed no statistical difference in alcoholic patients (table 2 and S2).”
4.2. “Besides, we found male gender, living in metropolis, and low family income was associated with higher rate alcoholic LC …”
There are only % data in table 1 for living area and income but no statistics whether these factors have a significant effect (as they are not included in table 2). So it’s not clear.
REPLY: We sincerely appreciate the positive feedback. We rephrase the sentence as follows:
“Besides, more male patients, living in metropolis, and low family income were found in the alcoholic LC group which were compatible with previous reports”
4.3. “We found that HE had the highest AHR” should be aHR
REPLY: We correct the error. Thank you.
4.3. “We found that HE had the highest AHR among the complications of LC.” But aHR for Ascites/Peritonitis in non-alcoholic LC is very high too, even higher than HE in this group. Please add and discuss as well.
REPLY: We sincerely appreciate the positive feedback. We rephrase the sentence as follows:
We found that HE had the highest aHR among the complications of alcoholic LC. Therefore, we should emphasize the prevention and management of HE in alcoholic LC patients. On the other hand, the highest aHR in non-alcoholic LC patients was ascites and peritonitis. Aggressive management of ascites by fluid or dietary sodium restriction and antibiotic prophylaxis for non-alcoholic LC patients with risk factors for spontaneous bacterial peritonitis may have some favourable impact on cirrhosis mortality.
|
Demographic data and comorbidities of male LC patients |
|||||
|
  |
Alcoholic Cirrhosis |
Non-alcoholic Cirrhosis |
  |
||
|
N=447 |
N=2699 |
  |
|||
|
Variable |
N |
% |
N |
% |
P value |
|
Age |
<0.001 |
||||
|
40-49 |
206 |
46.1 |
507 |
18.8 |
|
|
50-59 |
149 |
33.3 |
695 |
25.8 |
|
|
60-69 |
65 |
14.5 |
566 |
21.0 |
|
|
> 69 |
27 |
6.0 |
931 |
34.5 |
|
|
Residence |
0.004 |
||||
|
Metropolis |
232 |
51.9 |
1,526 |
56.5 |
|
|
General area |
196 |
43.8 |
1,123 |
41.6 |
|
|
Remote areas |
18 |
4.0 |
48 |
1.8 |
|
|
Income |
|
<0.001 |
|||
|
0-14009 |
184 |
41.2 |
1,310 |
48.5 |
|
|
14010-42030 |
240 |
53.7 |
1,138 |
42.2 |
|
|
> 42939 |
22 |
4.9 |
249 |
9.2 |
|
|
Comorbidity |
<0.001 |
||||
|
Yes |
311 |
69.6 |
2,083 |
77.2 |
|
|
No |
136 |
30.4 |
616 |
22.8 |
|
|
Coronary heart disease (CAD) |
57 |
12.8 |
614 |
22.7 |
<0.001 |
|
Cerebrovascular disease |
34 |
7.6 |
409 |
15.2 |
<0.001 |
|
Hemorrhage |
14 |
3.1 |
105 |
3.9 |
0.4363 |
|
Ischemia |
21 |
4.7 |
337 |
12.5 |
<0.001 |
|
Hypertension |
187 |
41.8 |
1,448 |
53.6 |
<0.001 |
|
Heart failure (HF) |
24 |
5.4 |
297 |
11.0 |
<0.001 |
|
Diabetes mellitus (DM) |
130 |
29.1 |
912 |
33.8 |
<0.001 |
|
Chronic kidney disease (CKD) |
70 |
15.7 |
569 |
21.1 |
<0.001 |
|
Hyperlipidemia |
114 |
25.5 |
617 |
22.9 |
0.2208 |
|
Chronic obstructive pulmonary disease (COPD) |
58 |
13.0 |
605 |
22.4 |
<0.001 |
Reviewer 2 Report
The authors do a good job of presenting a population-based cohort study to compare the survival between alcoholic and non-alcoholic LC. They effectively introduce the topic, provide relevant methods, and discuss the findings. Comparison with other studies provides additional perspective into the current population. The findings shed a good light on the role of sex and age, and other variables on mortality and comorbidities among alcoholic and non-alcoholic LC.
In Methods section, it should be made explicit that Kaplan-Meier method was used to conduct survival analysis and calculate the hazard ratios.
Author Response
We thank the editor and reviewers for the opportunity to revise manuscript (ID: ijerph-759609) entitled: “Comorbidities and Outcome of Alcoholic and Non-alcoholic Liver Cirrhosis in Taiwan – A Population-based Study." The manuscript has been corrected in accordance with the reviewer's comments. All of the reviewers’ and editors’ comments were included and responded point-by-point. The below responses were colored, and the changes made in the text and the tables are highlighted in red.
Response to Reviewer #2:
The authors do a good job of presenting a population-based cohort study to compare the survival between alcoholic and non-alcoholic LC. They effectively introduce the topic, provide relevant methods, and discuss the findings. Comparison with other studies provides additional perspective into the current population. The findings shed a good light on the role of sex and age, and other variables on mortality and comorbidities among alcoholic and non-alcoholic LC.
In Methods section, it should be made explicit that Kaplan-Meier method was used to conduct survival analysis and calculate the hazard ratios.
REPLY: We sincerely appreciate the positive feedback and valuable suggestions. We add this important information in the methods section as follows (highlighted in red):
2.4. Statistical and Data Analysis
To process the data from the NHIRD database, we used Microsoft SQL Server 2008 R2 (Microsoft Corporation, Redmond, WA, USA). All data were managed with the SPSS software 19.0 (SPSS, Inc., Chicago, IL). We used Chi-square test and analysis of variance, (ANOVA) to analyze the demographic data, concomitant comorbidities, and complications of cirrhosis. We used the Kaplan-Meier plot and Cox proportional hazards model to calculate the all-cause mortality hazard ratios (HRs) of comorbidities and complications of LC in all study subjects. The results are expressed in unadjusted and adjusted hazard ratios with 95% confidence intervals. The six-year cumulative survival plot was calculated using Cox regression model. A two-tailed p-value of less than or equal to 0.05 was considered statistically significant.
Reviewer 3 Report
Yang et a. compared clinical outcomes of alcoholic and non-alcoholic liver cirrhosis (LC) patients in a Taiwan population-based cohort study . The study revealed that alcoholic LC patients were younger, and had lower rates of concomitant comorbidities. However, the 6-year survival rates after diagnosis were not different between alcoholic and non-alcoholic LC. One of the most interesting results of this study is that hypertension and hyperlipidemia were associated with a lower risk of mortality.
Taiwan community is one of the viral hepatitis endemic areas. This study is of importance since it analysed the risk of concomitant comorbidities and complications of alcoholic and non-alcoholic LC patients in Taiwan. One of the major cavities of the study is that authors did not classify the cause of non-alcoholic liver patients (eg. viral hepatitis vs fatty liver).
Major points:
1. To better understand epidemiology of non-alcoholic liver disease in Taiwan patient cohort study, authors should classify non-alcoholic LC patients based on liver disease cause: eg. virus hepatitis, fatty liver or other autoimmune diseases.
2. How do authors reason that hypertension and hyperlipidemia are associated with a lower risk of mortality?
3. How much sex-based differences are associated with hypertension and hyperlidimia being a lower risk of mortality?
4. The authors described that the 6-year of survival rates after diagnosis were not different between alcoholic and non-alcoholic LC, what about the role of aging and other pathological hallmarks that are associated with age? Alcohol LC patients were younger compared to non-alcoholic LC patients.
Author Response
We thank the editor and reviewer for the opportunity to revise manuscript (ID: ijerph-759609) entitled: “Comorbidities and Outcome of Alcoholic and Non-alcoholic Liver Cirrhosis in Taiwan – A Population-based Study." The manuscript has been corrected in accordance with the reviewer's comments. All of the reviewers’ and editors’ comments were included and responded point-by-point. The below responses were colored, and the changes made in the text and the tables are highlighted in red.
Response to Reviewer #3:
Yang et a. compared clinical outcomes of alcoholic and non-alcoholic liver cirrhosis (LC) patients in a Taiwan population-based cohort study. The study revealed that alcoholic LC patients were younger, and had lower rates of concomitant comorbidities. However, the 6-year survival rates after diagnosis were not different between alcoholic and non-alcoholic LC. One of the most interesting results of this study is that hypertension and hyperlipidemia were associated with a lower risk of mortality.
Taiwan community is one of the viral hepatitis endemic areas. This study is of importance since it analyzed the risk of concomitant comorbidities and complications of alcoholic and non-alcoholic LC patients in Taiwan. One of the major cavities of the study is that authors did not classify the cause of non-alcoholic liver patients (eg. viral hepatitis vs fatty liver).
Major points:
- To better understand epidemiology of non-alcoholic liver disease in Taiwan patient cohort study, authors should classify non-alcoholic LC patients based on liver disease cause: eg. virus hepatitis, fatty liver or other autoimmune diseases.
We sincerely appreciate the positive feedback and suggestion. We add the etiologies of non-alcoholic LC data in table 1. The case number of autoimmune LC is zero in our 1,000,000 randomly sampled longitudinal health insurance database. The prevalence rate was reported to be 5.2 per 1,000,000 subjects in Taiwan (Koay LB et al. Dig. Dis. Sci. 2006). Therefore, we categorized all the patients without a diagnosis of HBV, HCV, or NAFLD into others.
- How do authors reason that hypertension and hyperlipidemia are associated with a lower risk of mortality?
REPLY: We sincerely appreciate the positive feedback. The cirrhosis-associated hemodynamic defects are low overall systemic vascular resistance, low renal blood flow, high arterial compliance, increased cardiac output, secondary activation of counter-regulatory systems, and resistance to vasopressors. Liver cirrhosis patients with raised arterial blood pressure constitute an effective counterbalance to this condition which may be associated with better survival than cirrhosis patients without hypertension. Impairment of lipid metabolism is common in chronic liver disease, especially in advanced liver cirrhosis patients. Privitera G., et al. conducted a systemic review and disclosed that liver cirrhosis patients with different etiologies have significant reductions in plasma lipid levels (including total cholesterol, HDL, LDL, and TG) (Privitera G., et al. Dig Dis Sci 2018). Low total cholesterol is correlated with a reduction in survival. Ghadir MR., et al. conducted a cross-sectional study and showed that serum total, LDL and HDL cholesterol levels in patients with cirrhosis had an inverse correlation with the severity of cirrhosis. (Ghadir MR., et al. Hepat Mon. 2010). Therefore, we believe that hyperlipidemia in cirrhosis patients suggests a better liver function and lower a lower risk of mortality.
- How much sex-based differences are associated with hypertension and hyperlipidemia being a lower risk of mortality?
REPLY: We sincerely appreciate the positive feedback. We conducted a further analysis of sex-based differences using male patients with comorbidity as reference group. When we analyzed the effect of different gender and hypertension on the mortality of LC patients, we used LC patients with hypertension as reference. In the alcoholic LC groups, we found no sex-difference associated with hypertension. In contrast, female without hypertension had a lower risk of mortality (HR = 0.792, P = 0.006) in the non-alcoholic LC group. We then analyzed the effect of different gender and hyperlipidemia on the mortality of LC patients, we used LC patients with hyperlipidemia as reference. In the alcoholic LC groups, we found both female with hyperlipidemia (HR = 8.146, P = 0.001) and male without hyperlipidemia (HR = 2.498, P < 0.001) had a higher risk of mortality. In contrast, we found both female without hyperlipidemia (HR = 1.626, P < 0.001) and male without hyperlipidemia (HR = 1.836, P < 0.001) had higher risk of mortality.
In conclusion, there are some sex-based differences associated with these two comorbidities in LC patients. However, we think that it would be hard to draw any conclusions due to the relatively small number of female cases in our study cohort. It is worthwhile to conduct further study to address this issue. Thank you.
- The authors described that the 6-year of survival rates after diagnosis were not different between alcoholic and non-alcoholic LC, what about the role of aging and other pathological hallmarks that are associated with age? Alcohol LC patients were younger compared to non-alcoholic LC patients.
REPLY: We sincerely appreciate the positive feedback. We agree with your points that evidence had shown that alcoholism or chronic alcohol consumption could cause both accelerated or premature aging. In addition, a systematic analysis of the global burden of alcohol use from 1990 to 2016 showed higher tuberculosis, road injuries, and self-harm related deaths in patients less than 50 years old. For populations aged 50 years and older, cancers accounted for a large proportion of total alcohol-attributable deaths (Collaborators, G.B.D.A. Lancet 2016). Unfortunately, we were not able to clarify the causes of death due to the limitations of the database.
Round 2
Reviewer 1 Report
Thank you very much for the replies and changes to the manuscript.